# Partial Information Decomposition and the Information Delta: A Geometric Unification Disentangling Non-Pairwise Information

**DOI:** 10.3390/e22121333

**Published:** 2020-11-24

**Authors:** James Kunert-Graf, Nikita Sakhanenko, David Galas

**Affiliations:** Pacific Northwest Research Institute, Seattle, WA 98122, USA; nsakhanenko@pnri.org (N.S.); dgalas@pnri.org (D.G.)

**Keywords:** partial information decomposition, information delta, synergy, co-information, non-pairwise dependence

## Abstract

Information theory provides robust measures of multivariable interdependence, but classically does little to characterize the multivariable relationships it detects. The Partial Information Decomposition (PID) characterizes the mutual information between variables by decomposing it into unique, redundant, and synergistic components. This has been usefully applied, particularly in neuroscience, but there is currently no generally accepted method for its computation. Independently, the Information Delta framework characterizes non-pairwise dependencies in genetic datasets. This framework has developed an intuitive geometric interpretation for how discrete functions encode information, but lacks some important generalizations. This paper shows that the PID and Delta frameworks are largely equivalent. We equate their key expressions, allowing for results in one framework to apply towards open questions in the other. For example, we find that the approach of Bertschinger et al. is useful for the open Information Delta question of how to deal with linkage disequilibrium. We also show how PID solutions can be mapped onto the space of delta measures. Using Bertschinger et al. as an example solution, we identify a specific plane in delta-space on which this approach’s optimization is constrained, and compute it for all possible three-variable discrete functions of a three-letter alphabet. This yields a clear geometric picture of how a given solution decomposes information.

## 1. Introduction

The variables in complex biological data frequently have nonlinear and non-pairwise dependency relationships. Understanding the functions and/or dysfunctions of biological systems requires understanding these complex interactions. How can we reliably detect interdependence within a set of variables, and how can we distinguish simple, pairwise dependencies from those which are fundamentally multivariable?

An analytical approach formulated by Williams and Beer frames these questions in terms of the Partial Information Decomposition (PID) [1]. The PID proposes to decompose the mutual information between a pair of source variables *X* and *Y* and a target variable *Z*, I(Z:X,Y), into four non-negative components:(1)$$\begin{array}{c} \begin{array}{cc} I(Z:X) & =U_{X}+R\\ I(Z:Y) & =U_{Y}+R\\ I(Z:X,Y) & =U_{X}+U_{Y}+R+S \end{array} \end{array}$$

The constituent terms proposed are: the *unique* informations, $U_{X}$ and $U_{Y}$, which represent the amounts of information about *Z* encoded by *X* alone and by *Y* alone; the *redundant* information, *R*, which is the information about *Z* encoded redundantly by both *X* and *Y*; and the *synergistic* information *S*, which is the information about *Z* contained in neither *X* or *Y* individually, but encoded by *X* and *Y* taken together. An illustration of this decomposition, the associated governing equations, and examples characterizing each type of information are all shown in Figure 1. It was shown that PID components can distinguish between dyadic and triadic relationships which no conventional Shannon information measure can distinguish [2].

The problem with this approach is that its governing equations form an underdetermined system, with only three equations relating the four components. To actually calculate the decomposition, an additional assumption must be made to provide an additional equation. Williams and Beer proposed a method for the calculation of *R* in their original paper, but this has since been shown to have some undesirable properties [1]. Much of the subsequent work in this domain consisted of attempts to define new relationships or formulae to calculate the components, as well as critiques of these proposed measures [3]. These proposed measures include (as an incomplete list): a measure based on information geometry [4]; an intersection information based on the Gács-Körner common information [5]; the minimum mutual information [6]; the pointwise common change in surprisal [7]; and the extractable shared information [8].

Another noteworthy putative solution is that of [9], which requires solving an optimization problem over a space *Q* of probability distributions, but is rigorous in that it directly follows from reasonable assumptions about the unique information. However, it is unclear how to sensibly generalize this approach to larger numbers of variables. Nonetheless, there has been considerable interest in using the PID approach to gather insights from real datasets, particularly within the neuroscience community [10,11,12,13,14,15].

Independently, an alternative approach to many of these same questions has been formulated focusing on devising new information theory-based measures of multivariable dependency. In genetics, non-pairwise epistatic effects are often crucially important in determining complex phenotypes, but traditional methods are sensitive only to pairwise relationships; thus there is particular interest in methods to identify the existence of synergistic dependencies within genetic datasets. Galas et al. [16,17] quantified the non-pairwise information between genetic loci and phenotype data with the *Delta measure*, Δ(Z:X,Y). Briefly, given a set of variables {X,Y,Z}, Δ(Z:X,Y) quantifies the change in co-information when considering the variables {X,Y,Z} as opposed to only {X,Y} (we hereafter denote Δ(Z:X,Y) as $\Delta_{Z}$, Δ(X:Y,Z) as $\Delta_{X}$, and so on). In its simplest application, the magnitudes of $\left\{\Delta_{X},\Delta_{Y},\Delta_{Z}\right\}$ can be used to detect and quantify non-pairwise interactions [16,17].

Recent work showed that the delta values encode considerable additional information about the dependency. Sakhanenko et al. [18] defined the normalized delta measures $\vec{\delta }=\left(\delta_{X},\delta_{Y},\delta_{Z}\right)$, which define an “information space”, and considered the δ-values of all possible discrete functions Z=f(X,Y). Fully mapping the specific example set of functions where {X,Y,Z} are all discrete variables with 3 possible values, they found that the 19,683 possible functional relationships Z=f(X,Y) mapped onto a highly structured plane in the space of normalized deltas (as shown in Figure 2). Different regions of this plane corresponded to qualitatively different types of functional relationships; in particular, completely pairwise functions such as Z=X and completely non-pairwise functions such as Z=XOR(X,Y) were mapped onto the extremes of the plane (see Figure 2; note that this paper defines “XOR” for a ternary alphabet as XOR(X,Y)≡(X+Y)mod3). Since discrete variables such as these occur naturally in genetics, this suggests that relationships between genetic variables may be usefully characterized by their $\vec{\delta }$-coordinates, with useful intuitive value. The difficulty of this in practice is that the coordinates are constrained to this plane only when *X* and *Y* are statistically independent, which is not the case in many real datasets, e.g., in genetic datasets in the presence of linkage disequilibrium.

In this paper, we show that the Partial Information Decomposition approach and Information Delta approach are largely equivalent, since their component variables can be directly related. The $\vec{\delta }$-coordinates can be written explicitly in terms of PID components, which leads us to an intuitive understanding of how $\vec{\delta }$-space encodes PID information by casting them into a geometric context. We then apply our results to two different approaches to solving the PID problem, first one from Bertschinger et al. [9] and then from Finn and Lizier [19]. We show that the sets of probability distributions, *Q*, used by Bertschinger can be mapped onto low-dimensional manifolds in $\vec{\delta }$-space, which intersect with the $\vec{\delta }$-plane of Figure 2. This approach is theoretically useful for the Delta information framework, since it factors out X,Y dependence in the data, thereby accounting for linkage disequilibrium between genetic variables. We suggest an approach for the analysis of genetic datasets which would return both the closest discrete function underlying the data and its PID in the Bertschinger solution, and which would require no further optimization after the initial construction of a solution library. This realization thus yields a low-dimensional geometric interpretation of this optimization problem, and we compute the solution for all possible three-variable discrete functions of alphabet size three. For these same functions, we then compute the PID components using the Pointwise PID approach of Finn and Lizier [19]. This visualization yields an immediate comparison of how each solution decomposes information. Since our derived relationship between the frameworks is general, it could be similarly applied to any putative PID solution as demonstrated here. Code to replicate these computations and the associated figures is freely available [20].

## 2. Background

### 2.1. Interaction Information and Multi-Information

An important body of background work, which served as a foundation for both the Information Decomposition and Information Delta approaches, involves the Interaction Information, II. II can be thought of as a multivariable extension of the mutual information [21]. Unlike the mutual information, however, the interaction information can assume negative values. What does it mean for the interaction information to be negative? It was once common to interpret I>0 as implying a synergistic interaction, and I<0 as implying a redundant interaction between the variables. As detailed in [1] and discussed in the following sections, this interpretation is mistaken. Interactions can be both partly synergistic and partly redundant, and the interaction information indicates the balance of these components.

For a set of variables $\nu_{n}=\left\{X_{1},...,X_{n}\right\}$, II can be defined as [22]:(2)$$II\left(\nu_{n}\right)=-\sum_{\tau_{i}\subseteq \nu_{n}}{\left(-1\right)}^{|\nu_{n}\left|-\right|\tau_{i}|}H\left(\tau_{i}\right)$$
where $|\nu_{n}|$ is the total number of variables in the set, and the sum is over all possible subsets $\tau_{i}$ (where $|\tau_{i}|$ is the total number of variables in each subset). $H(\tau_{i})$ is the joint entropy between the variables in subset $\tau_{i}$. The interaction information, II, is very similar to a measure called the co-information, CI [23]. These measures differ only by their sign: for an even number of variables they are identical (e.g., II(X,Y)=CI(X,Y)), and for an odd number of variables they are of opposite sign.
(3)$$CI\left(\nu_{n}\right)=-\sum_{\tau_{i}\subseteq \nu_{n}}{\left(-1\right)}^{|\tau_{i}|}H\left(\tau_{i}\right)={\left(-1\right)}^{|\nu_{n}|}II\left(\nu_{n}\right)$$

An additional, useful measure is the “multi-information”, Ω, introduced by Watanabe [24], sometimes called the “total correlation”, which represents the sum of all dependencies of variables and is zero only if all variables are independent. For a set of *n* variables $\nu_{n}=\left\{X_{i}\right\}$ it is defined as:(4)$$\Omega \left(\nu_{n}\right)=\sum_{i=1}^{n}H\left(X_{i}\right)-H\left(X_{1},\ldots X_{n}\right)$$

### 2.2. Information Decomposition

Consider a pair of “source variables” *X*,*Y* which determine the value of a “target variable” *Z*. Assume that we can measure the mutual information each source carries about a target, I(Z:X) and I(Z:Y) (which we abbreviate as $I_{Z:X}$ and $I_{Z:Y}$), as well as the mutual information between the joint distribution of {X,Y} and *Z*, I(Z:X,Y) (which we abbreviate as $I_{Z:XY}$). These mutual informations can be written in terms of the entropies (which we abbreviate using subscripts, e.g., $H\left(X,Y\right)\equiv H_{XY}$):(5)$$\begin{array}{c} \begin{array}{cc} I_{Z:X} & =H_{X}+H_{Z}-H_{XZ}\\ I_{Z:Y} & =H_{Y}+H_{Z}-H_{YZ}\\ I_{Z:XY} & =H_{XY}+H_{Z}-H_{XYZ} \end{array} \end{array}$$

These mutual informations can be decomposed into components which measure how much of each “type” of information they contain, as follows:(6)$$\begin{array}{c} \begin{array}{cc} I_{Z:XY} & =U_{X}+U_{Y}+R+S\\ I_{Z:X} & =U_{X}+R\\ I_{Z:Y} & =U_{Y}+R \end{array} \end{array}$$
where $U_{X}$ and $U_{Y}$ are the unique informations, *R* is the redundant information, and *S* is the synergistic information, as described previously in Section 1. This is an underdetermined system which requires an additional equation for the variables to render it solvable. Many of the current and previous efforts to define such an equation (for example, several proposals on how to directly compute the value of *R* from data), as well as the limitations of those efforts, have been nicely summarized in [3].

### 2.3. Solution from Bertschinger et al.

One solution to this problem came from Bertschinger et al. [9], who proposed that the unique information be approximated as:(7)$${\tilde{U}}_{X}=\min_{q\in Q}I\left(Y,Z|X\right)$$

Let Ψ be the set of all joint probability distributions of *X*, *Y*, and *Z*. Then we define *Q* as the set of all distributions, *q*, which have the same marginal probability distributions p(X=x,Z=z) and p(Y=y,Z=z) as our dataset. That is,
(8)Q={q∈Ψ|q(X=x,Y=y)=p(X=x,Y=y)∧q(Y=y,Z=z)=p(Y=y,Z=z)}

Please note that in [9], this set of probability distributions is denoted as $\Delta_{P}$, which we change here to *Q* to avoid notational confusion with the information deltas. Similarly, its elements are indicated by *Q* in the original paper. Here we indicate the distributions, elements of the set *Q*, by a lowercase *q* for consistency with our notation for probability distributions.

Put another way, we consider all possible probability distributions that maintain the marginals p(X=x,Z=z) and p(Y=y,Z=z) implied by our data. The relationship between *X* and *Y* (p(X=x,Y=y), and consequently the joint distribution p(X=x,Y=y,Z=z)) is allowed to vary. The minimization criterion is perhaps more intuitive when written, equivalently, as:(9)$${\tilde{U}}_{X}=\min_{q\in Q}I\left(Y,Z|X\right)=\min_{q\in Q}\left[II\left(X,Y,Z\right)-II\left(y,z\right)\right]$$

Thus, the unique information ${\tilde{U}}_{X}$ can be thought of as the smallest possible increase in the interaction information when the variable *X* is added to the set {Y,Z}. For example, if there exists a probability distribution in *Q* for which II(Y,Z)=II(X,Y,Z), then the addition of *X* adds no unique information about *Z* and ${\tilde{U}}_{X}$. The core assumption of this approach is that the unique and redundant informations depend simply upon the marginal distributions p(X=x,Z=z) and p(Y=y,Z=z). This solution is rigorous in the sense that the result follows directly from this assumption without any ad-hoc assumptions for how the components are related.

### 2.4. Information Deltas and Their Geometry

Consider a set of three variables $\nu_{n}=\left\{X,Y,Z\right\}$. Using Equation (Equation 3), we can write the co-information in terms of the entropies:(10)$$CI\left(X,Y,Z\right)=H_{X}+H_{Y}+H_{Z}-H_{XY}-H_{XZ}-H_{YZ}+H_{XYZ}$$

The differential interaction information $\Delta_{Z}$ is the change in the interaction information when a given variable *Z* is added to the set. This can be written in terms of CI and then the conditional mutual information:(11)$$\Delta_{Z}\left(\nu_{n}\right)=CI\left(X,Y,Z\right)-CI\left(X,Y\right)=-I\left(X,Y|Z\right)$$

These measures can be normalized by the multi-information for the three variables, Ω(X,Y,Z) (which we abbreviate as $\Omega_{XYZ}$), which by Equation (Equation 4) we can write in terms of the entropies as:(12)$$\Omega_{XYZ}=H_{X}+H_{Y}+H_{Z}-H_{XYZ}$$
The normalized measures are then:(13)$$\delta_{X}=-\Delta_{X}/\Omega_{XYZ},\;\delta_{Y}=-\Delta_{Y}/\Omega_{XYZ},\;\delta_{Z}=-\Delta_{Z}/\Omega_{XYZ}$$

If *Z* is a function of *X* and *Y*, and if *X* and *Y* are i.i.d., then $\vec{\delta }=\left(\delta_{X},\delta_{Y},\delta_{Z}\right)$ lies within a highly structured plane, where different regions of the plane correspond to qualitatively different types of interactions. Figure 2 shows the mapping of all possible functions onto this highly structured plane.

The normalized deltas can be expressed as:(14)$$\begin{array}{c} \begin{array}{cc} \delta_{X} & =1-\frac{I_{X:Y}+I_{Z:X}}{\Omega_{XYZ}}=\frac{-H_{X}+H_{XY}+H_{XZ}-H_{XYZ}}{\Omega_{XYZ}}\\ \delta_{Y} & =1-\frac{I_{X:Y}+I_{Z:Y}}{\Omega_{XYZ}}=\frac{-H_{Y}+H_{XY}+H_{YZ}-H_{XYZ}}{\Omega_{XYZ}}\\ \delta_{Z} & =1-\frac{I_{Z:X}+I_{Z:Y}}{\Omega_{XYZ}}=\frac{-H_{Z}+H_{XZ}+H_{YZ}-H_{XYZ}}{\Omega_{XYZ}} \end{array} \end{array}$$

The normalized deltas can also be written in terms of joint mutual informations, as follows:(15)$$\begin{array}{c} \begin{array}{cc} \delta_{X} & =\frac{1}{\Omega_{XYZ}}\left(-H_{X}+H_{XY}+H_{XZ}-H_{XYZ}\right)\\ \; & =\frac{1}{\Omega_{XYZ}}\left(-H_{X}+H_{XY}+H_{XZ}-H_{XYZ}+\left(H_{X}+H_{Y}+H_{Z}-H_{XYZ}\right)-\Omega_{XYZ}\right)\\ \; & =\frac{1}{\Omega_{XYZ}}\left(H_{Y}+H_{Z}+H_{XY}+H_{XZ}-2H_{XYZ}-\Omega_{XYZ}\right)\\ \; & =\frac{1}{\Omega_{XYZ}}\left(\left(H_{Z}+H_{XY}-H_{XYZ}\right)+\left(H_{Y}+H_{XZ}-H_{XYZ}\right)-\Omega_{XYZ}\right)\\ \; & =\frac{I_{Z:XY}+I_{Y:XZ}}{\Omega_{XYZ}}-1 \end{array} \end{array}$$

We can write all normalized deltas in this form:(16)$$\begin{array}{c} \begin{array}{cc} \delta_{X} & =\frac{I_{Z:XY}+I_{Y:XZ}}{\Omega_{XYZ}}-1\\ \delta_{Y} & =\frac{I_{Z:XY}+I_{X:YZ}}{\Omega_{XYZ}}-1\\ \delta_{Z} & =\frac{I_{Y:XZ}+I_{X:YZ}}{\Omega_{XYZ}}-1 \end{array} \end{array}$$

By inverting previous equations, we can then write:
(17a)$$\begin{array}{cc} I_{Z:XY} & =\frac{\Omega_{XYZ}}{2}\left(\delta_{X}+\delta_{Y}-\delta_{Z}+1\right) \end{array}$$
(17b)$$\begin{array}{cc} I_{Z:X} & =\frac{\Omega_{XYZ}}{2}\left(-\delta_{X}-\delta_{Z}+\delta_{Y}+1\right) \end{array}$$
(17c)$$\begin{array}{cc} I_{Z:Y} & =\frac{\Omega_{XYZ}}{2}\left(-\delta_{Y}-\delta_{Z}+\delta_{X}+1\right) \end{array}$$
Specifically, Equation Set (Equation 16) can be inverted to yield Equation ([Disp-formula FD17a-entropy-22-01333]), and Equation Set (Equation 14) can be inverted to yield Equations (17b) and (17c).

## 3. PID Mapped into Information Deltas

### 3.1. Information Decomposition in Terms of Deltas

With Equations (Equation 6) and (17), we can equate the expressions for the mutual informations in their delta and information decomposition forms:(18)$$\begin{array}{c} \begin{array}{cc} \frac{\Omega_{XYZ}}{2}\left(+\delta_{X}+\delta_{Y}-\delta_{Z}+1\right) & =R+U_{X}+U_{Y}+S\\ \frac{\Omega_{XYZ}}{2}\left(-\delta_{X}+\delta_{Y}-\delta_{Z}+1\right) & =R+U_{X}\\ \frac{\Omega_{XYZ}}{2}\left(+\delta_{X}-\delta_{Y}-\delta_{Z}+1\right) & =R+U_{Y} \end{array} \end{array}$$

From the above relations we can derive:(19)$$S-R=\frac{\Omega_{XYZ}}{2}\left(\delta_{X}+\delta_{Y}+\delta_{Z}-1\right)$$

In other words, the difference between the synergy and the redundancy increases as we get farther from the origin in δ-space. Also:(20)$$U_{X}-U_{Y}=\Omega_{XYZ}\left(\delta_{Y}-\delta_{X}\right)$$
so the distance from the diagonal in the $\left(\delta_{X},\delta_{Y}\right)$-plane is proportional to the difference between the unique informations. These striking relationships are visualized in Figure 3.

### 3.2. Relationship between Diagonal and Interaction Information

Considering again Equation (Equation 19) and using Equation (Equation 13), we can write:(21)$$\begin{array}{c} \begin{array}{cc} S-R & =\frac{\Omega_{XYZ}}{2}\left(\delta_{X}+\delta_{Y}+\delta_{Z}-1\right)\\ \; & =-\frac{1}{2}\left(\Delta_{X}+\Delta_{Y}+\Delta_{Z}+\Omega_{XYZ}\right)\\ \; & =-\frac{1}{2}(\left(H_{X}-H_{XY}-H_{XZ}+H_{XYZ}\right)+\left(H_{Y}-H_{XY}-H_{YZ}+H_{XYZ}\right)\\ \; & \;+\left(H_{Z}-H_{XZ}-H_{YZ}+H_{XYZ}\right)+\left(H_{X}+H_{Y}+H_{Z}-H_{XYZ}\right))\\ \; & =-(H_{X}+H_{Y}+H_{Z}-H_{XY}-H_{XZ}-H_{YZ}+H_{XYZ})\\ \; & =II(X,Y,Z) \end{array} \end{array}$$
where II(X,Y,Z) is the interaction information between the variables. This replicates the important result that II(X,Y,Z)=S−R from the original Williams and Beer paper [1].

### 3.3. The Function Plane

When the variables are related by a discrete function (as defined in [18]), and *X* and *Y* are i.i.d., the function will lie on a plane defined by:(22)$$\delta_{Z}=\delta_{X}+\delta_{Y}-1$$
Thus, the distance *d* of a coordinate above the plane is given by
(23)$$d=\delta_{Z}-\delta_{X}-\delta_{Y}+1=-\left(\delta_{X}+\delta_{Y}-\delta_{Z}+1\right)+2$$
And so from Equation (Equation 18):(24)$$\frac{\Omega }{2}\left(2-d\right)=R+U_{X}+U_{Y}+S$$

## 4. Solving the PID on the Function Plane

### 4.1. Transforming Probability Tensors within *Q*

As noted previously, there is no generally accepted solution for completing and computing the set of PID equations. Our results connecting the PID to the information deltas have therefore, up to this point, been agnostic on this question. All equations in the previous section follow from the basic PID formulation, and the delta coordinate equations. This means they are true for any putative solution, but also brings us no closer to an actual solution to the PID problem; we can still only compute the differences between PID components. We therefore now extend our analysis by using the solution of Bertschinger et al. [9] to fully compute the PID for the functions in Figure 2. We wish to emphasize, however, that the following approach could be used equally well to gain a geometric interpretation of any alternate solution to the PID.

Consider a probability tensor for an alphabet size of N:(25)$$P_{N}=\left[\left[\begin{array}{ccc} p_{111} & \cdots & p_{1N1}\\ \vdots & \ddots & \vdots \\ p_{N11} & \cdots & p_{NN1} \end{array}\right],\;...\;,\left[\begin{array}{ccc} p_{11N} & \cdots & p_{1NN}\\ \vdots & \ddots & \vdots \\ p_{N1N} & \cdots & p_{NNN} \end{array}\right]\right]$$
where we use the notation $p_{ijk}=p\left(X=i,Y=j,Z=k\right)$. What transformations are permissible that will preserve the distribution within the set *Q* (as defined in Equation (Equation 8))? Please note that we can obtain the marginal distributions simply by summing over the appropriate tensor index. For example, summing along the first index yields the marginal distribution p(Y=y,Z=z). To stay in *Q*, then, we require that the sums along the first and second indices both remain constant.

For an alphabet size of N=2, we can parameterize the set of all possible transformations quite simply:(26)$$P_{2}=\left[\left[\begin{array}{cc} p_{111}+\alpha & p_{121}-\alpha \\ p_{211}-\alpha & p_{221}+\alpha \end{array}\right]\;,\;\left[\begin{array}{cc} p_{111}+\beta & p_{121}-\beta \\ p_{211}-\beta & p_{221}+\beta \end{array}\right]\right]$$

All possible changes to each layer of the tensor can be captured with a single parameter. For example, increasing $p_{111}$ will require that $p_{121}$ and $p_{211}$ be decreased by the same amount, as the row and column sums must remain constant (which, in turn, determines $p_{221}$). Each layer can be modified independently, and thus the second layer has an independent parameter.

For a given probability tensor with N=2, then, the probability tensor for any distribution in *Q* can be fully parameterized with two parameters, and thus the corresponding coordinates in delta-space are at most two-dimensional. In practice, we find that N=2 functions have delta-coordinates that are restricted to a one-dimensional manifold.

Consider, for example, the AND function:(27)$$P_{2}=\left[\left[\begin{array}{cc} 1 & 1\\ 1 & 0 \end{array}\right]\;,\;\left[\begin{array}{cc} 0 & 0\\ 0 & 1 \end{array}\right]\right]$$

We can describe all possible perturbations which remain in *Q* by the parameterization:(28)$$P_{2}=\left[\left[\begin{array}{cc} 1/4+\alpha & 1/4-\alpha \\ 1/4-\alpha & 0+\alpha \end{array}\right]\;,\;\left[\begin{array}{cc} 0+\beta & 0-\beta \\ 0-\beta & 1/4+\beta \end{array}\right]\right]$$
However, it can be seen that we must have β=0, as all probabilities must remain in the range p∈[0,1]. The parameter α, on the other hand, can fall within the range α∈[0,1/4]. Since all possible perturbations can be captured by varying a single parameter, *Q* must therefore be mapped to a one-dimensional manifold in δ-space.

The layers of a probability tensor become significantly harder to parameterize for N=3. Consider a single layer of a probability tensor:(29)$$\left[\begin{array}{ccc} p_{111} & p_{121} & p_{131}\\ p_{211} & p_{221} & p_{231}\\ p_{311} & p_{321} & p_{331} \end{array}\right]\equiv \left[\begin{array}{ccc} a & b & c\\ e & d & f\\ g & h & i \end{array}\right]$$
The permissible transformations to this layer can be parameterized by:(30)$$\left[\begin{array}{ccc} a+\alpha & b+\beta & c-\alpha -\beta \\ e+\gamma & d+\delta & f-\gamma -\delta \\ g-\alpha -\gamma & h-\beta -\delta & i+\alpha +\beta +\gamma +\delta \end{array}\right]$$
subject to the constraints that:(31)$$\left\{\begin{array}{c} 0\leq (a+\alpha )\leq 1\\ 0\leq (b+\beta )\leq 1\\ 0\leq (e+\gamma )\leq 1\\ 0\leq (d+\delta )\leq 1\\ 0\leq (c-\alpha -\beta )\leq 1\\ 0\leq (f-\gamma -\delta )\leq 1\\ 0\leq (g-\alpha -\gamma )\leq 1\\ 0\leq (h-\beta -\delta )\leq 1\\ 0\leq (i+\alpha +\beta +\gamma +\delta )\leq 1 \end{array}\right.$$

Clearly, these relations are too complicated to lend any immediate insight into the problem. However, it is a simple matter to use the above inequalities to calculate permissible values of parameters (α,β,γ,δ) and to plot out the corresponding delta coordinates. This is done for randomly generated sample functions in Figure 4. In this case, the delta coordinates have a complex distribution but are nonetheless restricted to a plane in delta-space (the vertically oriented red plane in Figure 5).

### 4.2. δ-Coordinates in *Q* Are Always Restricted to a Plane

In the N=2 case, delta-coordinates were parameterized by a single variable such that they must be restricted onto a line. In the N=3 example, they are restricted onto a plane. Will larger alphabets map *Q* onto a three-dimensional volume? If not, is it possible to get a non-planar two-dimensional manifold, or are coordinates always restricted to a plane? We will now prove that *Q* is always constrained to a plane, regardless of the alphabet size.

**Lemma** **1.**
*In any set Q as defined previously, the following entropies remain constant: all individual entropies $H_{X}$, $H_{Y}$ and $H_{Z}$; the joint, 2-variable entropies containing Z, namely $H_{XZ}$ and $H_{YZ}$. The only entropies which vary within a particular Q then are $H_{XY}$ and $H_{XYZ}$.*


**Proof.** The definition of *Q* preserves the marginal distributions by construction. $H_{XZ}$ and $H_{YZ}$ being constant is a trivial consequence of holding p(X=x,Z=z) and p(Y=y,Z=z) constant, which is the condition defining *Q*. From these constant marginal distributions, we can calculate the distributions p(X=x), p(Y=y) and p(Z=z), which are therefore also constant, as are their corresponding entropies. □

Only two entropic quantities vary between the different distributions in *Q*. By considering just their effect on the delta coordinates, we can now show the following:

**Theorem** **1.**
*In any set Q of distributions with equal marginal distributions p(X=x,Z=z) and p(Y=y,Z=z), the delta-coordinates $\left(\delta_{X},\delta_{Y},\delta_{Z}\right)$ will be restricted to a plane. This is true for any alphabet size.*


**Proof.** We begin by making several notational definitions to simplify the algebra which follows, first from the joint entropies which vary within *Q*:
$$\begin{array}{cc} d & \equiv H_{XYZ}-H_{XY}\\ h & \equiv H_{XYZ} \end{array}$$
We then define quantities which collect the constant entropy terms:
$$\begin{array}{cc} c_{1} & \equiv H_{X}+H_{Y}+H_{Z}\\ c_{2} & \equiv -H_{X}+H_{XZ}\\ c_{3} & \equiv -H_{Y}+H_{YZ}\\ c_{4} & \equiv -H_{Z}+H_{XZ}+H_{YZ} \end{array}$$
In terms of these quantities we can now write the normalized delta coordinates as follows:
$$\begin{array}{cc} \delta_{X} & =\frac{c_{2}-d}{c_{1}-h}\\ \delta_{Y} & =\frac{c_{3}-d}{c_{1}-h}\\ \delta_{Z} & =\frac{c_{4}-h}{c_{1}-h} \end{array}$$
Solving for *d* in the $\delta_{X}$ and $\delta_{Y}$ equations yields:
$$\delta_{Y}\left(c_{1}-h\right)-c_{3}=\delta_{X}\left(c_{1}-h\right)-c_{2}$$
And the $\delta_{Z}$ equation allows us to solve for *h*:
$$h=\frac{c_{4}-c_{1}\delta_{Z}}{1-\delta_{Z}}\Rightarrow \left(c_{1}-h\right)=\frac{c_{1}-c_{4}}{1-\delta_{Z}}$$
Plugging this into the equation above yields an equation which simplifies to:
(32)$$\left(c_{1}-c_{4}\right)\left(\delta_{X}-\delta_{Y}\right)+\left(c_{3}-c_{2}\right)\left(1-\delta_{Z}\right)=0$$
Since $c_{1},c_{2},c_{3}$ and $c_{4}$ are all constant over *Q*, this defines a plane in $\delta_{X},\delta_{Y},\delta_{Z}$ space. □

Equation (Equation 32) not only shows that the points in *Q* are bound to a plane, but it also implies that this plane always contains the line defined by $\delta_{X}=\delta_{Y}$ and $\delta_{Z}=1$. Therefore for any function in Figure 2, we can trivially compute the plane in which the corresponding *Q* is contained.

### 4.3. PID Calculation for All Functions

For the set of probability distributions *Q*, Bertschinger et al. [9] provide the following estimators for the PID components:(33)$$\begin{array}{c} \begin{array}{cc} \; & {\tilde{U}}_{X}=\min_{q\in Q}\Omega_{XYZ}\delta_{Y}\\ \; & {\tilde{U}}_{Y}=\min_{q\in Q}\Omega_{XYZ}\delta_{X}\\ \; & \tilde{R}=\max_{q\in Q}CI\left(X,Y,Z\right)\\ \; & \tilde{S}=I_{Z:XY}-\min_{q\in Q}I_{Z:XY} \end{array} \end{array}$$

If we numerically compute the set *Q* for a given function *f* (i.e., by generating a distribution such as the one shown in Figure 4 via the parameterization of Equation (Equation 30)), these estimators are trivially consistent. Figure 6 shows the computed values of the PID components for all of the functions shown in Figure 2. There is a clear geometric interpretation here: Functions in the lower left/right corners consist almost entirely of ${\tilde{U}}_{X}$ and ${\tilde{U}}_{Y}$, respectively. Functions approaching the top corner become increasingly synergistic with a higher proportion of *S*. Functions are most redundant towards the lower center of the plane, though no single function is primarily *R*.

### 4.4. Alternate Solutions: Pointwise PID

The Pointwise Partial Information Decomposition (PPID) of Finn and Lizier [19] is an alternate approach to solving the PID problem. It is motivated by the fact that the entropy and mutual information can be expressed as the expectation value of pointwise quantities, which measure the information content of a single event. For example, the event $\left(X,Z\right)=\left(x_{1},z_{1}\right)$ has the associated pointwise mutual information:(34)$$i\left(x_{1}:z_{1}\right)=\log\frac{p(X=x_{1}|Z=z_{1})}{p(X=x_{1})}$$
and the overall mutual information between the two variables is the expectation value of this pointwise quantity, taken over all possible events. It is important to note that while the overall mutual information is non-negative, the pointwise mutual information *can* be negative. Finn and Lizier decompose this pointwise quantity into two non-negative components, the “specificity” $i^{+}\left(x_{1}\to z_{1}\right)$ and “ambiguity” $i^{-}\left(x_{1}\to z_{1}\right)$, and argue that:(35)$$\begin{array}{c} \begin{array}{cc} i(x_{1}:z_{1}) & =i^{+}\left(x_{1}\to z_{1}\right)-i^{-}\left(x_{1}\to z_{1}\right)\\ i^{+}\left(x_{1}\to z_{1}\right) & =h\left(x_{1}\right)=-\log p\left(X=x_{1}\right)\\ i^{-}\left(x_{1}\to z_{1}\right) & =h\left(x_{1}|z_{1}\right)=-\log p\left(X=x_{1}|Z=z_{1}\right) \end{array} \end{array}$$

They similarly decompose the redundancy *R* into a pointwise specific redundancy $r^{+}$ and pointwise specific ambiguity $r^{-}$, and argue for the following definitions:(36)$$\begin{array}{c} \begin{array}{cc} r_{min}^{+}\left(a_{1},...,a_{k}\to z\right) & =\min_{a_{i}}i^{+}\left(a_{i}\to z\right)\\ r_{min}^{-}\left(a_{1},...,a_{k}\to z\right) & =\min_{a_{i}}i^{-}\left(a_{i}\to z\right) \end{array} \end{array}$$
where $\left\{a_{i}\right\}$ are the values of each of the source variables in a particular realization (e.g., if we have two source variables X,Y predicting *Z*, then the event $\left(X,Y,Z\right)=\left(x_{1},y_{2},z_{3}\right)$ has $\left\{a_{i}\right\}=\left\{x_{1},y_{2}\right\}$). The expectation value of the difference of these quantities then yields the redundancy:(37)$$R=\langle r_{min}^{+}-r_{min}^{-}\rangle$$
from which the rest of the PID components follow. See [19] for a full discussion of motivations and Axioms which these definitions satisfy (including a discussion of the relationship between this formulation and that of Bertschinger et al. [9], and how the many aspects of [19] are arguably pointwise adaptations of the assumptions in [9]).

One consequence of this approach is that the PID components are no longer non-negative. There is extensive discussion of the interpretation of this in [19], but one example, RdnErr, is particularly informative. In our probability tensor notation, we can write this function as:(38)$$P_{R}dnErr=\left[\left[\begin{array}{cc} 3/8 & 1/8\\ 0 & 0 \end{array}\right]\;,\;\left[\begin{array}{cc} 0 & 0\\ 1/8 & 3/8 \end{array}\right]\right]$$
This can be interpreted as follows: *X* is always equal to *Z*. *Y* is *usually* equal to *Z*, but occasionally (with probability 1/4) makes an error. What should we expect the PID components to be, in this case? The PPID yields $\left(R,U_{x},U_{y},S\right)=\left(1,0,-0.81,0.81\right)$, which implies the following interpretation: the information about *Z* is encoded *redundantly* by both *X* and *Y*, but *Y* carries unique *mis*information about *Z* due to its tendency to make errors. If all components were strictly positive, we would likely draw a different conclusion: both *X* and *Y* encode *some* information about *Z*, with *X* encoding *additional* unique information. In this way, different solutions will lead to slightly different interpretations about the nature of the relationship between the variables.

In Figure 7, we compute the PPID for all functions Z=f(X,Y) and map them onto $\vec{\delta }$-space, just as we did in Figure 6. Comparing Figure 6 and Figure 7 immediately highlights key differences in how each method decomposes information. For example: in Figure 6, the top corner is purely synergistic, the lower-left corner has information solely in *X*, and the lower-right corner has information solely in *Y*; in Figure 7, the top corner has zero redundancy, the lower-left corner has misinformation in *Y*, and the lower-right corner has misinformation in *X*.

It is not our goal here to argue which result is *more* correct. Instead, we wish to highlight how comparing Figure 6 and Figure 7 readily yields subtle insights into how the two approaches differ in decomposing information. It also yields immediate insights into the subtleties of how we might interpret coordinates in $\vec{\delta }$-space.

## 5. Conclusions

The key overall result of this paper is that the PID problem can be mapped directly into the the previously defined “information landscape” represented by the “delta space” of [18]. This theoretical framework is simple and has a geometric interpretation which was well worked out previously. The simple set of relations between the frameworks, as explicated in Equation (Equation 18) and visualized in Figure 3, anticipates a much deeper set of geometric constraints.

We build upon this general relationship using the solution of Bertschinger et al. [9]. Using this solution, we parameterize the permissible transformations to a discrete function to numerically generate the distribution set *Q*, and prove in Theorem 1 that this set is mapped onto a plane in delta-space. The optimization problem defined by this approach is cast in terms of our variables in Equation (Equation 33), and the various extrema can be extracted directly from our parameterization and mapping procedure. Code which replicates these computations and generates the figures within this paper is freely available[20].

These results suggest the following approach for computation of the PID components, if using the solution from [9], and given the added assumption that there is some function Z=f(X,Y) which best approximates the relationship between variables. The steps are these:Construct a library (set) of distributions $\left\{Q_{1},Q_{2},\ldots ,Q_{N}\right\}$ for all functions, $f_{i}\left(X,Y\right)$. Specifically, record the δ-coordinates spanned by each distribution (e.g., as plotted in Figure 4) along with the corresponding function and its PID component values.For a set of variables in data for which we wish to find the decomposition, compute its δ-coordinates and then match them to the closest $Q_{i}$. This will then immediately yield the corresponding function and PID components.

If this approach proves to be practical, it would have several clear advantages. First, the computational cost of the library construction would only need to be done once, and not need to be repeated for any subsequent analysis. The cost of the library construction is itself quite tractable (for example, exactly this computation was done to generate Figure 6). Second, this solves an open problem in the use of Information Deltas for which the source variables are not independent, for example, in applications to genetics in the presence of linkage disequilibrium. Specifically, this approach relaxes the common assumption in [18] that *X* and *Y* must be statistically independent.

The practical application of this approach to data analysis requires further development, which is beyond the scope of this paper. Specifically, the actual data will contain noise such that the computed δ-coordinates will not lie perfectly within any distribution of *Q* set. The naïve approach of simply taking the closest $Q_{i}$ may therefore be insufficient in general. Future work will characterize the response of δ-coordinates to various levels of noise within the data, to enable the computation of $p(Q_{i}|\vec{\delta },\alpha )$ (i.e., the probability that variables belong to the set $Q_{i}$ given their observed coordinates $\vec{\delta }$ and some noise level α).

Future work will extend the approach of $\vec{\delta }$ to larger sets of variables, so as to fully characterize a higher-dimensional $\vec{\delta }$-space and its relationship to the PID. Much of the complexity of each framework is contained in considering these higher-order relationships. Future work will also consider additional solutions to the PID problem beyond the solutions of [9,19] considered here. All equations in Section 3 are general and agnostic to the precise solution used for the actual PID computation, and it should be straightforward to generate figures similar to Figure 6 for different solutions to show how they differ in mapping information components onto the function plane. This will provide interpretable geometric comparisons between solutions and also immediately highlight all functions for which results offer differing interpretations, as seen in Figure 6 and Figure 7. We anticipate that this direct comparison of how different solutions map the information content of discrete functions will provide a powerful visual tool for understanding the differing consequences of putative solutions, and thus our unification of these frameworks will be useful in resolving the open question of how best to compute the PID.

## Figures and Tables

**Figure 1 entropy-22-01333-f001:**
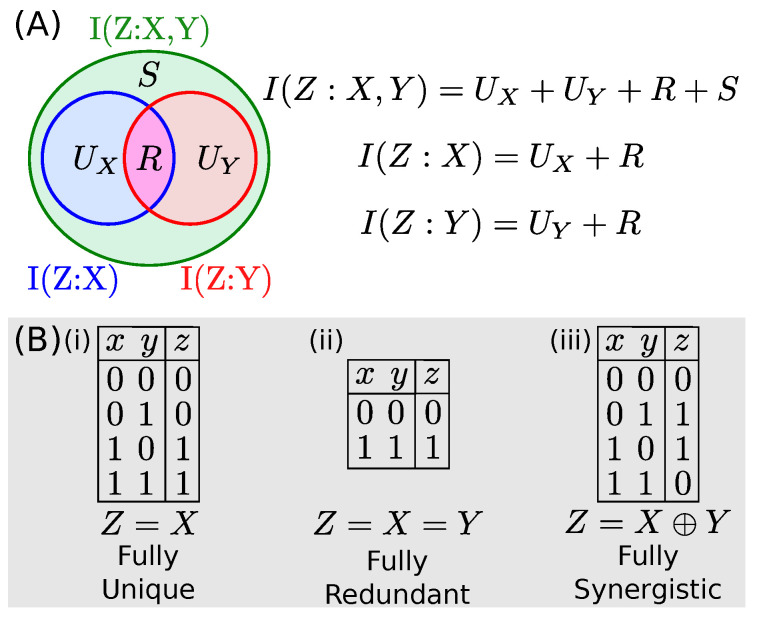
(**A**) Visualization of the Information Decomposition (adapted from [3]) and its governing equations. The system is underdetermined. (**B**) Sample binary datasets which contain only one type of information. For (i), where Z=X, *X* contains all information about *Z* and *Y* is irrelevant, such $U_{X}$ is equal to the total information and all other terms are zero. For (ii), where Z=X=Y, *X* and *Y* are always identical and thus the information is fully redundant. For (iii), where *Z* is the XOR function of *X* and *Y*, both *X* and *Y* are independent of *Z*, but fully determine its value when taken jointly.

**Figure 2 entropy-22-01333-f002:**
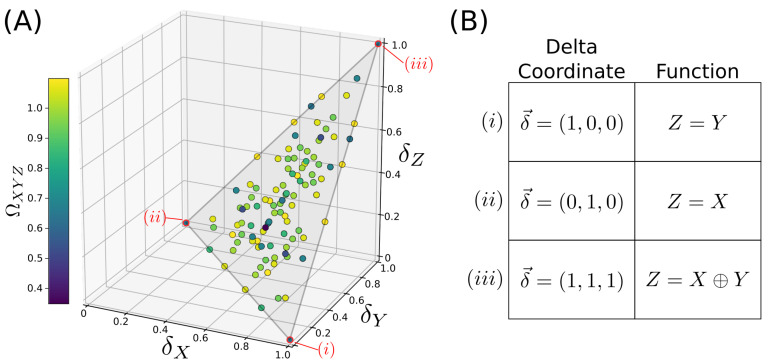
A geometric interpretation of the Information Deltas, as developed in [18]. (**A**) Consider functions where each variable has an alphabet size of three possible values. There are 19,683 possible functions f(X,Y). If the variables *X* and *Y* are independent, these functions map onto 105 unique points (function families) within a plane in δ-space. (**B**) Sample functions and their mappings onto δ-space. Functions with a full pairwise dependence on *X* or *Y* map to opposite lower corners, whereas the fully synergistic XOR (i.e., the XOR-like ternary extension XOR(X,Y)≡(X+Y)mod3) is mapped to the uppermost corner.

**Figure 3 entropy-22-01333-f003:**
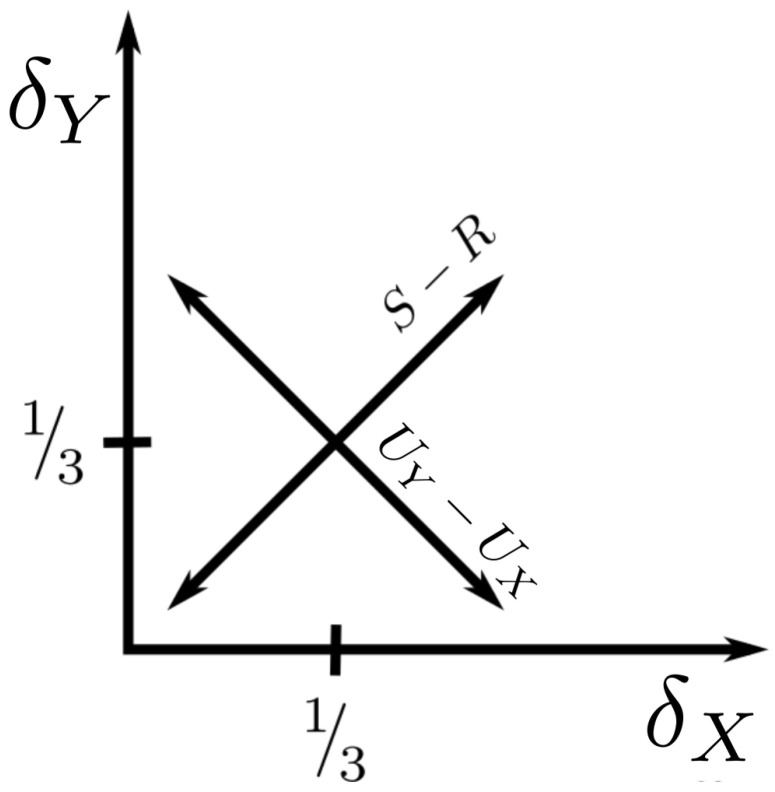
As shown in Equations (Equation 23) and (Equation 24), the δ-space encodes the balance of synergy/redundancy along one diagonal, and the balance of unique information in each source along the other.

**Figure 4 entropy-22-01333-f004:**
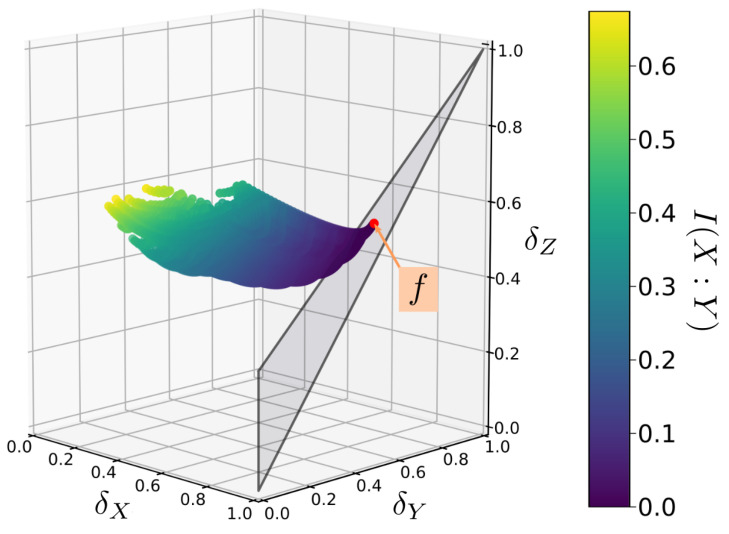
An example mapping of the Bertschinger set *Q* to δ-space for a randomly chosen function *f*. A set *Q* consists of all probability distributions p(X=x,Y=y,Z=z) that share the same marginal distributions p(X=x,Z=z) and p(Y=y,Z=z). Each *Q* maps onto a set of points with a complex distribution, but which is constrained to a simple plane in δ-space.

**Figure 5 entropy-22-01333-f005:**
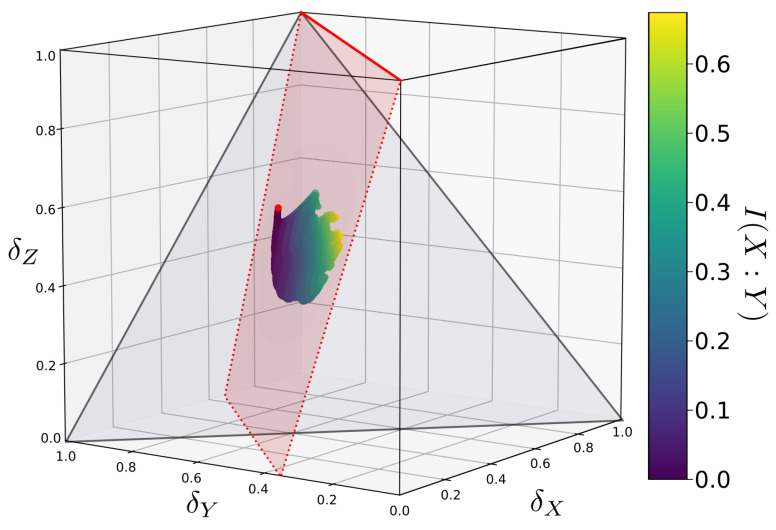
The same function’s *Q* mapped onto δ-space as in Figure 4, viewed from a different angle. *Q* is constrained to a plane in δ-space. This plane, highlighted in red, contains the δ-coordinates of the function *f* (indicated by the red dot) as well as the line$\left(\delta_{X}=\delta_{Y},\delta_{Z}=1\right)$ (indicated by the solid red line).

**Figure 6 entropy-22-01333-f006:**
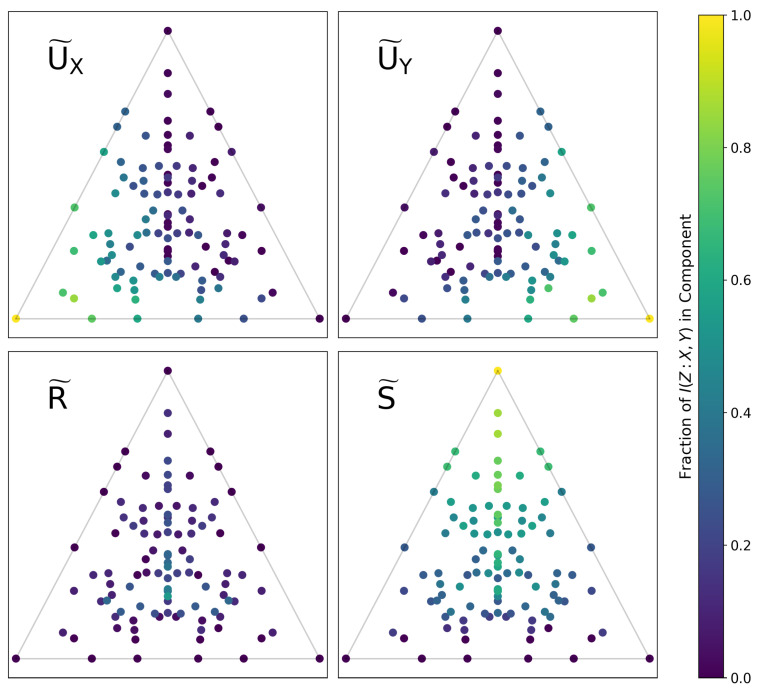
All functions Z=f(X,Y) (with alphabet sizes of 3) mapped onto a plane in δ-space, as in Figure 2. Each function is colored by the fraction of the total information in each PID component, as computed using the solution of [9]. There is a clear geometric structure to the decomposition which matches the previously discussed intuition about δ-space.

**Figure 7 entropy-22-01333-f007:**
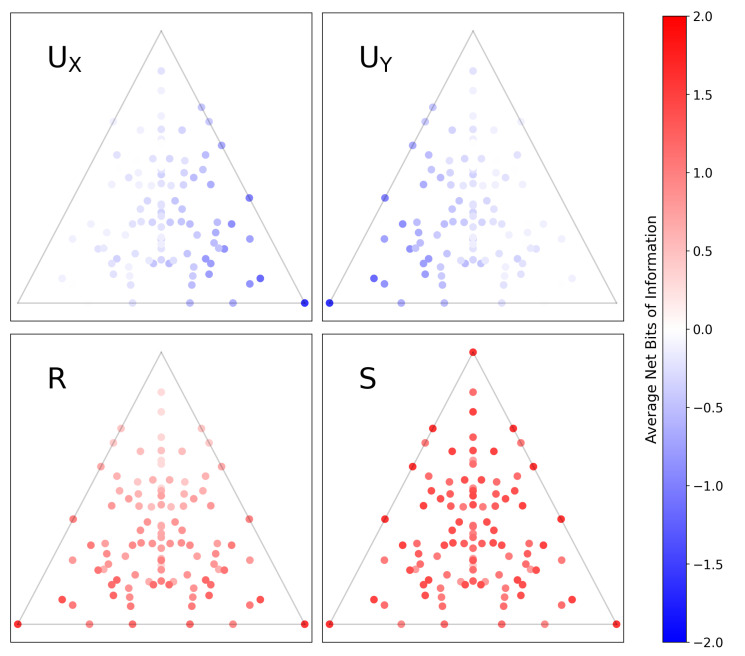
The same set of all 3-letter functions Z=f(X,Y) mapped onto a plane in δ-space, as in Figure 6. The colorscale shows the amount of information in each component, now computed using the pointwise solution of Finn and Lizier [19]. In this formulation, the PID components are the average difference between two subcomponents, the specificity and ambiguity, and can be negative when the latter exceeds the former. Visualizing this solution immediately highlights the differences in how it decomposes the information of functions and leads to an alternate interpretation of δ-space.

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
