# Peer review of "Partial Information Decomposition and the Information Delta: A Geometric Unification Disentangling Non-Pairwise Information"

_entropy, 2020, doi:10.3390/e22121333_

Round 1

Reviewer 1 Report

Kunert-Graf and colleagues present the mathematical link between PID problems and the information delta-method. Specifically, they present an example linking the optimization space Q to the BROJA-PID. Here, the BROJA-PID measure defines a plane in the Delta-space that intersects at specific points with the delta plane spanned by the set of standrad functions in the same space. Equations and proofs to us seem to present no issues and we appreciate the geometrical perspective introduced by the authors.

Major:

1) We have struggled to find a definition for the XOR function on ternary variables (i.e. variables with alphabets of size three) used in Fig. 2. We think that “XOR” is closely connected to binary Boolean logic for most readers, including us. We suspect that the function from Fig. 2 is actually the arithmetic sum followed by the modulo(3) operation, but remain unsure, since we could not find a proper definition. We have also asked our colleagues in mathematics but did not get a conclusive answer. Thus. , either a definition of that function or a proper reference should be given.

2) The two input variable case (three variables in total) hides most of the appeal and also the intricacies of the PID problem. If the Delta-method should be of use to the PID community, we strongly suggest to present the connection of PID and Information-Delta method at least for one PID measure taking three inputs (four variables in total) and yielding a proper lattice of terms. We would suggest Lizier’s pointwise measure, but I_min from the original Williams and Beer PID paper would also be OK. Without doing so we actually find the results of reduced use for the PID community. Especially, because no deeper insights on the Bertschinger measure seem to have been drawn from the use of the Information-Delta method, and from the fact that a plane is found for the Bertschinger measure in the information-delta space. In case we have overlooked these, we would suggest to stress them more.

3) The ‘recursion relation’ for the II (equation 2) is actually undefined - because it does not yield a definition for a conditional II, and no definition for a conditional II is given anywhere else in the mansucript. ( Remember that Entropies and conditional entropies are different objects from the point of view of mathematical structure; the same may or may not hold for a conditional II and the reader cannot tell. Since this recursion relation is unnecessary it could potentially be simply dropped.

4) Please use a notation that makes the distinction between probability distributions and probabilities and between random variables and their realization clear.

5) The notation will not hold work for three input variables, or four variables in total (i.e. decomposing I(V:X,Y,Z). Please use a standard MI-separator symbol, i.e. the colon “:” I(V:X,Y,Z) or the semicolon “;” I(V;X,Y,Z). We may suggest to stick with the notation introduced by Bertschinger et al. for this case, who use a colon to separate left and right arguments in the mutual information and PID terms, the semicolon to separate different collections of variables, between which redundancies are computed, and comma to separate the variables. For example the redudancy between the coalition of X and Y and the solitary variable Z about V is written as: I_shd(V:X,Y;Z). But any other notation that avoids confusion between left and right arguments of the information expression and the list of collections entering as the right argument would acceptable.

6) The introduction for our taste too much mixes the PID problem structure (as introduced by Williams and Beer) with a specific implementation that has some peculiarities (from Bertschinger), the most importnat one being limited to just two inputs. Possibly this could be disentangled.

Minor:

1) Place colorbars in all figures that need them (esp. Fig. 5).

2) Eq. 2 and 3 have minor typesetting issues (parenthesis don’t match → subscript level/main text line)

3) There are some more well-cited studies in the neuroscience community that used PID; in afield still in its infancy, it may be good to mention most of the available studies to give the reader an impression about the usefulness and importance of this nascent field.

- Kay, J. W., Ince, R. A., Dering, B., & Phillips, W. A. (2017). Partial and entropic information decompositions of a neuronal modulatory interaction. Entropy, 19(11), 560.

- Wibral, M., Finn, C., Wollstadt, P., Lizier, J. T., & Priesemann, V. (2017). Quantifying information modification in developing neural networks via partial information decomposition. Entropy, 19(9), 494.

Author Response

Thank you for your comments. Please see the attached response.

Reviewer 2 Report

Dear authors

thanks for submitting this work. I am happy to see that there are different approaches, developed from different backgrounds, and with different applications in mind, that then converge to similar concepts.

Indeed, even in the closer circle of people working on Partial Information Decomposition, I am routinely amazed (and sometimes frustrated) when I realize that different names are used for similar concepts, or similar names are used for different concepts. And what makes things worse is that whether concepts are or not the same things is sometimes a subjective account.

Now, in this light, I welcome your effort to document, with analyses, definitions, and publicly available code, the analogies between information delta and partial information decomposition, and I arbitrarily and unsolicitedly welcome you to the PID field (or vice-versa I am happy to belong as of today to the information delta field).

It would be nice if you could briefly comment on some aspects with which we have all been struggling so far

  • definition of how we can complement Shannon's formulation in the PID framework: the MMI (Minimum MI) Barrett, A. B. (2015). Exploration of synergistic and redundant information sharing in static and dynamical Gaussian systems. Physical Review E91(5), 052802. https://journals.aps.org/pre/abstract/10.1103/PhysRevE.91.052802. And other definitions, apart the one by Bertschinger et al. that you cite, namely
    • Harder, M., Salge, C., & Polani, D. (2013). Bivariate measure of redundant information. Physical Review E87(1), 012130.
    • Griffith, V., Chong, E. K., James, R. G., Ellison, C. J., & Crutchfield, J. P. (2014). Intersection information based on common randomness. Entropy16(4), 1985-2000.
    • Ince, R. A. (2017). Measuring multivariate redundant information with pointwise common change in surprisal. Entropy19(7), 318.
    • James, R. G., & Crutchfield, J. P. (2017). Multivariate dependence beyond Shannon information. Entropy19(10), 531.
  • Some criticism, like for example
    • James, R. G., Barnett, N., & Crutchfield, J. P. (2016). Information flows? A critique of transfer entropies. Physical review letters116(23), 238701.

Thanks for listening

Author Response

(The authors gave the same response as above.)
